# Ovarian Stimulation with FSH Alone versus FSH plus a GnRH Antagonist for Elective Freezing in an Oocyte Donor/Recipient Programme: A Protocol for a Pilot Multicenter Observational Study

**DOI:** 10.3390/jcm12072743

**Published:** 2023-04-06

**Authors:** Ioannis E. Messinis, Christina I. Messini, Evangelos G. Papanikolaou, Evangelos Makrakis, Dimitrios Loutradis, Nikolaos Christoforidis, Theodosis Arkoulis, Georgios Anifandis, Alexandros Daponte, Charalampos Siristatidis

**Affiliations:** 1Department of Obstetrics and Gynecology, Faculty of Medicine, School of Health Sciences, University of Thessaly, 415 00 Larissa, Greece; messinis@med.uth.gr (I.E.M.); messini.uth@gmail.com (C.I.M.); ganif@med.uth.gr (G.A.); daponte@med.uth.gr (A.D.); 2Assisting Nature IVF Unit, Giannou Kranidiotis, & Avenue 2, 570 01 Thessaloniki, Greece; papanikolaou@assistingnature.gr; 3HYGEIA IVF EMBRYOGENESIS A.R.T. Unit, Fleming 15, 151 23 Athens, Greece; evmakrakis@gmail.com; 4Institute of Fertility, Pasteur 15, Mavili Square, 115 21 Athens, Greece; loutradi@otenet.gr; 5Embryolab IVF Unit, Leoforos Ethnikis Antistaseos 175, 551 34 Thessaloniki, Greece; n.christoforidis@embryolab.eu; 6Mitosis IVF Centre, Sachtouri 24, 185 37 Pireaus, Greece; tark25@otenet.gr; 7Assisted Reproduction Unit, Second Department of Obstetrics and Gynecology, “Aretaieion” Hospital, Medical School, National and Kapodistrian University of Athens, 76 Vas. Sofias Av., 115 28 Athens, Greece

**Keywords:** ovarian stimulation protocol, GnRH agonist, GnRH antagonist, in vitro fertilization, assisted reproductive techniques, oocyte donation, LH surge

## Abstract

Preliminary data have shown that it is possible to attempt in vitro fertilization (IVF) treatment in fresh cycles without the use of a gonadotropin-releasing hormone (GnRH) antagonist or any other medication to prevent the luteinizing hormone (LH) surge during ovarian stimulation. To date, there is no information on this topic in the context of a prospective controlled trial. However, as prevention of the LH surge is an established procedure in fresh cycles, the question is whether such a study can be performed in frozen cycles. We aim to perform a pilot study in order to compare the efficacy of a protocol using FSH alone with that of a protocol using follicle-stimulating hormone (FSH) plus a GnRH antagonist for controlled ovarian hyperstimulation (COH) in cycles of elective freezing in the context of a donor/recipient program. This is a seven-center, two-arm prospective pilot cohort study conducted at the respective Assisted Reproductive Units in Greece. The hypothesis to be tested is that an ovarian stimulation protocol that includes FSH alone without any LH surge prevention regimens is not inferior to a protocol including FSH plus a GnRH antagonist in terms of the clinical outcome in a donor/recipient model. The results of the present study are expected to show whether the addition of the GnRH antagonist is necessary in terms of the frequency of LH secretory peaks and progesterone elevations >1 ng/mL during the administration of the GnRH antagonist according to the adopted frequency of blood sampling in all Units.

## 1. Introduction

It is known that during controlled ovarian stimulation (COH) with follicle-stimulating hormone (FSH) for in vitro fertilization (IVF), the positive feedback mechanism can be activated by estradiol, before the follicle is fully matured, leading to the occurrence of a premature luteinizing hormone (LH) surge, which is the main responsible factor for premature luteinization. To prevent a premature LH surge, gonadotropin-releasing hormone (GnRH) agonists and antagonists are routinely used in COH protocols with FSH [1].

Existing data have shown high efficacy of the GnRH agonists in the prevention of LH surges [2,3], whereas the currently used GnRH antagonists have been less effective in controlling pituitary LH secretion. In specific contexts, various studies have reported multiple LH secretory peaks during the administration of a GnRH antagonist at a frequency ranging from 8–35%, with a concomitant rise in progesterone levels in several cases [4,5,6,7,8]. In addition, the GnRH antagonist, ganirelix, at a dose of 0.25 mg per day for 5 days, has been unable to block the positive feedback effect of exogenous estradiol in experiments performed in normal female volunteers during the follicular phase of their menstrual cycle [9]. Furthermore, a previous study has shown that after the administration of one dose of 0.25 mg of the GnRH antagonist, cetrorelix, secretory LH pulses were eliminated for about 6 h, but then a gradual recovery of the pituitary was observed, with a re-establishment of the normal pattern of LH pulses during the last 6 h before the administration of the next dose [10]. This indicates that with the current protocols of daily administration of the GnRH antagonists during COH, there is a period between two successive doses of the antagonist during which the anterior pituitary is not protected against the high levels of the circulating estradiol, which can activate the positive feedback mechanism and the occurrence of LH secretory peaks. Finally, GnRH antagonists do not appear to reduce the basal levels of LH when administered either during COH or in the follicular phase of a normal menstrual cycle [6,9,11]. Notably, during COH with FSH, the administration of a GnRH antagonist begins on day 6 of the treatment cycle, i.e., at a time when blood levels of LH are already significantly reduced due to the increased activity of the negative feedback mechanism from the rising estradiol levels; thus, at that point, the addition of the GnRH antagonist cannot cause a further decrease [6,9] and, probably, is therefore not necessary.

Despite the above-mentioned “weaknesses” of the GnRH antagonists, no significant difference in live birth rate has been shown between COH protocols for IVF using agonists and those using antagonists [12]. This could be interpreted as indicating that any premature rise in LH or even progesterone levels during COH is of negligible value for a successful final outcome. In other words, failure to administer a drug to block the LH surge in IVF cycles may not have a significant impact on the clinical outcome. Preliminary data have shown that it is possible to attempt an IVF treatment in fresh cycles without the use of a GnRH antagonist or any other medication to prevent a LH surge during COH [13]. It was interesting that women receiving a GnRH antagonist had LH levels during stimulation similar or even significantly higher than women without an antagonist [13]. To date, there is no information on this topic in the context of a prospective controlled trial. However, as the prevention of the LH surge is an established procedure in fresh cycles, the question is whether such a study can be performed in frozen cycles.

Recently, progesterone-primed protocols have been introduced in cycles with “freeze all” embryos [14]. The results have shown that the live birth rate following frozen/thawed embryo transfer does not differ significantly between protocols with progesterone and those with GnRH antagonists or agonists, while there is no difference in the congenital malformation risk either [15,16]. This could lead to the safe conclusion that, in the case of elective freezing, GnRH analogues are not necessary for LH surge prevention, while GnRH agonists could be used for triggering the final oocyte maturation. On the other hand, premature luteinization has been shown to reduce the chance of pregnancy in fresh but not in frozen cycles, thus suggesting that progesterone adversely affects the endometrium, which is not a concern in frozen cycles [17]. If so, then it is questionable whether exogenous progesterone is also necessary to prevent the LH surge in elective freezing cycles [18].

The aim of this pilot study is to compare the efficacy of a protocol using FSH alone with that of a protocol using FSH plus a GnRH antagonist for COH in cycles of elective freezing in the context of a donor/recipient program. The hypothesis to be tested is that a COH protocol that includes FSH alone without any LH surge prevention regimens is not inferior to a protocol including FSH plus a GnRH antagonist in terms of clinical outcomes.

## 2. Materials and Methods

### 2.1. Study Design

This is a seven-center, two-arm prospective pilot cohort study to be conducted at the Assisted Reproductive Units of:a.The Department of Obstetrics and Gynecology, Faculty of Medicine, School of Health Sciences, University of Thessaly, Larissa, Greece;b.The Assisted Reproduction Unit, Second Department of Obstetrics and Gynecology, “Aretaieion” Hospital, Medical School, National and Kapodistrian University of Athens, Athens, Greece;c.HYGEIA IVF—Embryogenesis A.R.T. Unit, Athens, Greece;d.The Institute of Fertility, Athens, Greece;e.The Embryolab IVF Unit, Thessaloniki, Greece;f.The Mitosis IVF Centre, Pireaus, Greece;g.The Assisting Nature IVF Unit, Thessaloniki, Greece.

The first two University Units will offer scientific and academic support, while in the rest, the five private Units, the recruitment of the study subjects will take place.

Patients will be recruited from the acceptance of the protocol from the Scientific Committees of all Units. The study protocol has been registered prospectively in clinicaltrials.gov (NCT05759871).

### 2.2. Eligibility Criteria

The inclusion criteria of the study population are: healthy women between 21–32 years old with a BMI of 21 to 29 kg/m^2^ who wish to donate their oocytes; normal ovarian reserve tests; normal menstrual cycles of 26–32 days; absence of any hormonal treatment during the last three months before entering the study and absence of coagulation and/or autoimmune disorders.

The exclusion criteria include: use of other protocols towards oocyte retrieval, such as natural, or modified natural cycles; poor ovarian response according to the Bologna criteria [19]; history of endocrine or metabolic disorders, ovarian cystectomy or oophorectomy; women with a diagnosis of polycystic ovary syndrome; clinical and/or laboratory markers of hereditary or acquired thrombophilia that complied to the standard protocols of each Unit and non-hormonal medication for a serious medical condition.

### 2.3. Participants—Recruitment

For this study, female donors participating in a donor/recipient program will be recruited following a detailed explanation of the study protocol. The selection will be based on the eligibility criteria. All participants will sign a written informed consent before entering the trial. The women will be divided into two groups, i.e., group 1 (study) and group 2 (control). The allocation of patients will be achieved using a 1:1 proportional pattern, depending on the random order in which they enter each Unit. Notably, both patients and gynecologists will be aware of the study protocol used.

### 2.4. Interventions

#### 2.4.1. Controlled Ovarian Hyperstimulation

All women will undergo ovarian stimulation with urinary FSH (Meriofert, IBSA Pharmaceutici, Lodi, Italy) starting on day 2 of the cycle. The starting dose of FSH will be 300 IU s.c. for the first 4 days adjusted thereafter according to the ovarian response. Monitoring of treatment will be performed via the measurement of serum estradiol concentrations and transvaginal ultrasound scans of the ovaries. Triggering of final follicle/oocyte maturation will be performed via the injection of a GnRH agonist at the single dose of 200 µg s.c. when at least 3 follicles greater than 17 mm in diameter are seen via ultrasound. Women in group 1 will receive only FSH, while women in group 2 will additionally receive a GnRH antagonist from day 6 of FSH treatment at a dose of 0.25 mg per day until the triggering day. The GnRH antagonist in group 2 will be injected each time immediately after the injection of FSH. Monitoring via ultrasound scans will begin on cycle day 7 and continue every day or every other day, depending on the examiner’s evaluation. In every scan, all follicles >10 mm in diameter (mean of two dimensions) will be recorded. From all women (both groups), blood samples (5 mL) will be obtained before the FSH injection (between 08.00 and 09.00 h) on cycle days 2 and 7 and then every other day. The last blood sample will be taken on the day of triggering regardless of the sampling frequency adopted. All blood samples will be centrifuged, and an aliquot will be stored at −20° until assayed.

#### 2.4.2. Endometrium Preparation

In recipients, there will be no fresh but only frozen/thawed embryo transfer. For this purpose, their endometrium will be prepared by administering hormone replacement therapy. The women will receive oral estradiol valerate as follows: 2 mg on cycle days 2–3, 4 mg on days 4–5 and 6 mg from day 6 onwards. Once the endometrium, at least 10 days after starting estradiol treatment, becomes thicker than 8 mm in diameter as assessed via a transvaginal ultrasound scan, micronized progesterone (Utrogestan caps 200 mg, Faran A.B.E.E., Athens, Greece) will be given vaginally at the daily dose of 400 mg (200 mg two times per day) or progesterone (Vasclor vag. Gel 8%, 22.5 G, Verisfield U.K. Ltd., Athens, Greece) once daily depending on the participating unit’s protocol. Estradiol valerate administration will continue at a dose of 4 mg per day. The maximum number of embryos to be transferred will be 2 per patient.

### 2.5. Intended Comparisons and Outcomes

The primary outcome measures will be the frequency of LH secretory peaks during the administration of the GnRH antagonist according to the adopted frequency of blood sampling. A secretory peak of LH is defined as the increase at levels ≥ 10 IU/L. Progesterone elevation > 1 ng/mL any time during the ovarian stimulation period will be also evaluated. Secondary outcome measures will be ongoing clinical pregnancy (fetal heartbeat at ultrasound) and miscarriage rates. Other prespecified outcome parameters included the total dose of gonadotropins (FSH), the endometrial thickness at day of triggering, number of retrieved cumulus oocytes complexes (COCs), MII oocytes, transferred embryos, day of embryo transfer, biochemical pregnancy (positive β-hCG), multiple pregnancy, ectopic pregnancy and live birth rates. Maternal side effects from the use of both drugs will also be monitored, including ovarian hyperstimulation syndrome. Embryo quality for embryo transfer or cryopreservation will be assessed according to morphological criteria [20].

Our primary analysis will be performed to provide a direct comparison of the study and control groups. We will also carry out logistic regression analysis, inserting variables, including demographic and IVF cycle characteristics, in order to reveal potential independent predictors that could potentially contribute to the change in the primary outcomes and live birth rate.

### 2.6. Calculation of Sample Size

The present trial will be based on a consecutive series of patients. The sample size will be calculated as follows: If there is a true difference in favor of the experimental treatment of 5% (20% vs. 15%), then 160 patients (80 per group in case of 1:1 enrollment) are required to be 80% sure that the upper limit of a one-sided 95% confidence interval (or equivalently a 90% two-sided confidence interval) will exclude a difference in favor of the standard/control group of more than 10%. According to a pre-decided pattern from all authors, a pilot study will proceed the full procedure; in that, a total of 50 study subjects will constitute the first cohort in a ratio of 1:1 in both groups.

### 2.7. Statistical Analysis

All analyses will be carried out using the Intention to Treat (ITT) principle. The outcomes of the variables included will be expressed as the median [range]. We will assess the normality of the distributions with Kolmogorov–Smirnov’s test and graphical methods. We will select the non-parametric Mann–Whitney U-test to compare continuous variables because both aforementioned tests were suggestive of an abnormal distribution. Categorical variables will be assessed with the chi-square and Fisher’s exact test. All tests will be two-sided. Logistic regression analysis will be performed to determine the potential predictors of clinical pregnancy, miscarriage and live birth rates using the Enter method.

## 3. Discussion

The results of the present study are expected to show whether an ovarian stimulation protocol with FSH alone has similar clinical efficacy after frozen/thawed embryo transfer to a protocol using a GnRH antagonist in addition to FSH in the context of a donor/recipient program.

COH is the most crucial stage for the success of IVF as it leads to the necessary creation of oocytes available for retrieval and subsequently the creation of the selected embryo(s) for endometrial transfer. Among the three phases of the procedure—multi-follicular development, pituitary suppression towards the prevention of the LH surge and premature ovulation and triggering of oocyte maturation [21]—the second appears to be theoretically necessary. Premature ovulation is rare and also possible with GnRH antagonist use, even in in progestin-primed ovarian stimulation and natural cycles of IVF. The rationale behind its obligatory inhibition during an IVF cycle is that it impedes a woman’s chance to reach her full oocyte yield potential and subsequently the success of assisted reproduction. However, the possibility of premature ovulation is rather unlikely, because a follicle usually ruptures when it is mature, i.e., when it exceeds 16 mm in diameter in the presence of a fully expressed LH surge [22]. In the case of a premature LH surge, on the one hand, the follicles usually have not fully matured, yet on the other hand, the LH surge is markedly attenuated and even less than a third of the normal amplitude. Data in mice have shown that for the follicle to rupture, the LH surge must be at least 85% of the normal size [23].

Despite the fact that during ovarian stimulation with FSH the threshold for the positive feedback is exceeded early in the cycle, an endogenous LH surge has been reported in about 20% of stimulated cycles [2]. This relatively low incidence of a spontaneous LH surge during ovarian stimulation with FSH alone has been attributed to the overproduction by the ovaries of a substance named gonadotrophin surge-attenuating factor (GnSAF), which antagonizes the sensitizing effect of estradiol on the response of the pituitary to GnRH, thereby weakening or disabling the positive feedback mechanism [24,25,26]. Thus, GnSAF causes a marked attenuation or even complete blockade of the LH surge [27,28]. However, the inability to predict in which cycles a premature LH surge will occur during ovarian stimulation is the reason for the use of the relevant drugs in all women, although only about two in ten will benefit.

The rationale behind trying to prevent the LH surge during ovarian stimulation is to prevent the associated premature luteinization. Even a markedly attenuated LH surge can lead to premature luteinization [29]. Evidence has been provided that premature luteinization can cause advanced maturation of the endometrium or glandular–stromal dyssynchrony, which adversely affects the clinical outcome [30]. This would be the main obstacle to undertaking COH without GnRH analogues in fresh cycles. However, with the recent development of vitrification, which makes it possible to freeze all embryos with high success, this obstacle may be overcome. It is important to note that there is no consensus regarding the cut off level of serum progesterone that defines luteinization. Different studies have investigated the impact of various levels, from 0.8 ng/mL up to 3 ng/mL, on the clinical outcome during ART [17]. For the purpose of the present study, in cycles with a secretory LH peak, a progesterone cut off level will be considered the value of 1 ng/mL, as previously reported [4,5,6].

The results of the present study may suggest a different approach to ovarian stimulation in fresh and frozen cycles. Some differences between these two kinds of cycles have already been reported. For example, in fresh cycles, it has been shown that the live birth rate increases in proportion to the number of oocytes retrieved up to a number of 15, after which there is a plateau and then a significant decrease [31,32,33]. In contrast, in frozen cycles, the live birth rate after frozen/thawed embryo transfer steadily increases up to a number of >25 oocytes retrieved [33]. Furthermore, in fresh cycles, the live birth rate decreases significantly with increasing the daily or total dose of FSH regardless of the number of oocytes retrieved [34], which is not the case with frozen cycles [35,36]. Although these data suggest that there may be a negative impact of the high dose of FSH on the endometrium, further data in a donor/recipient model with fresh ET have indicated that it is the oocyte rather than the endometrium that is adversely affected [37]. It is still, however, a matter of debate as to whether the risk of aneuploidy increases with the increase in the total dose of FSH [38,39,40].

The current trend in cycles of elective freezing is an intense ovarian stimulation with the retrieval of a relatively high number of oocytes that ensures a sufficient number of frozen/thawed embryo transfers in subsequent cycles, and this method appears to demonstrate a major advantage, leading to high cumulative live birth rates [41]. At the same time, the administration of a GnRH agonist for triggering final oocyte maturation eliminates the risk of OHSS. Conversely, in fresh cycles, obtaining more than 15 oocytes and especially >18 greatly increases the risk of OHSS [33].

Avoiding the administration of any drugs to inhibit the LH surge in cycles of elective freezing could benefit women by ensuring lower treatment costs, although there are no studies on a cost–benefit analysis, friendlier procedures without many injections and participants with no exposure to any further drug exposure. The hypothesis that was tested is based on the strong arguments mentioned above, and after obtaining the results, it appears to have a high possibility of being true.

Finally, we admit that this is an observational study that will carry the associated biases a priori; as a result, no definite conclusion can be drawn. In contrast, we believe that this is a new theory that has to be proven step-by-step. We fully acknowledge the anxiety of both patients and clinicians for premature LH surges and ovulation; especially for the later, we came across a lot of negative beliefs and fear when we tried to recruit IVF centres for participation. Our further—and initial—aim, though, remains to conduct a proper RCT to prove or disprove the outlined theory, but to do so with some preliminary data in our hands.

## Data Availability

Not applicable.

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
