# Peer review of "Ovarian Stimulation with FSH Alone versus FSH plus a GnRH Antagonist for Elective Freezing in an Oocyte Donor/Recipient Programme: A Protocol for a Pilot Multicenter Observational Study"

_jcm, 2023, doi:10.3390/jcm12072743_

Round 1
Reviewer 1 Report
This is a protocol for a proposed pilot and full trial comparing COS protocols, one using GnRH antagonist to prevent LH surge to one using only FSH and not using any GnRH analogs to prevent the surge, specifically in egg donor cycles. In general this is presented in a clear manner with corresponding rationale, and I wish you well in this endeavor. I do have a few questions about the protocol as below.
1. Can you be more specific when you discuss how women will be allocated to each treatment arm? I assume they will be randomly assigned? What randomization method will you be using? This is obviously more complex with a multicenter trial.
2. Presumably some portion of women in the no-antagonist arm will ovulate prior to their planned oocyte retrieval. How will you handle their management? I see the LH and progesterone levels will be run at a later time, but after ovulation follicle sizes will decrease and estradiol will drop.
3. I see you will be using intent to treat, so just confirming that the above women who do get canceled will be included in the analysis?
4. There is a statement in your discussion I find confusing, might be reading it incorrectly. Line 216-219. You appear to be stating that pituitary suppression to prevent the LH surge and premature ovulation is absolutely necessary, but you are testing with this protocol whether this is necessary or now so this seems contradictory.
5. There is much focus in your discussion on premature luteinization being the possible negative effect of early LH surge. I agree that this won't be a concern with egg donor cycles, but premature ovulation will still be a risk for these cycles and may bear some discussion as well.
Author Response
Comments
This is a protocol for a proposed pilot and full trial comparing COS protocols, one using GnRH antagonist to prevent LH surge to one using only FSH and not using any GnRH analogs to prevent the surge, specifically in egg donor cycles. In general this is presented in a clear manner with corresponding rationale, and I wish you well in this endeavor. I do have a few questions about the protocol as below.
A: Thank you for your time spent to carefully assess our paper and the valuable comments.
- Can you be more specific when you discuss how women will be allocated to each treatment arm? I assume they will be randomly assigned? What randomization method will you be using? This is obviously more complex with a multicenter trial.
A: Thank you for your comment. This is not an RCT, where meticulous randomization process is obligatory. This is an observational cohort study. To further clarify allocation in the two arms, some sentences have been added. It now reads:” The allocation of patients will be achieved using a 1:1 proportional pattern, depending on the random order in which they enter each Unit. Notably, both patients and gynecologists will be aware of the study protocol used.”.
- Presumably some portion of women in the no-antagonist arm will ovulate prior to their planned oocyte retrieval. How will you handle their management? I see the LH and progesterone levels will be run at a later time, but after ovulation follicle sizes will decrease and estradiol will drop.
A: Thank you for your comment. This is a vey interesting point. As we state in the manuscript (new addition in the discussion section), the rate of pre oocyte retrieval ovulation is expected to be low. In addition, in order for our theory to be proved, we will not proceed to cancellation of the cycle. Another point that we have in mind, is that premature ovulation -if happens- will be in some and not all the growing follicles, and at a diameter of less than 16mm; we strongly believe that, even if this is the case, this will not hamper the rest of the follicles and they will be able to contain mature oocytes. However, it is important to emphasize that follicles with a diameter of <16 mm are quite difficult to rupture (see answer point 4).
- I see you will be using intent to treat, so just confirming that the above women who do get canceled will be included in the analysis?
A: Yes, the value of our study will be based on an intention to treat final analysis and not per protocol. This means that parameters will be analyzed per initial number of women and not excluding those who will have a premature ovulation or LH surge.
- There is a statement in your discussion I find confusing, might be reading it incorrectly. Line 216-219. You appear to be stating that pituitary suppression to prevent the LH surge and premature ovulation is absolutely necessary, but you are testing with this protocol whether this is necessary or now so this seems contradictory.
A: We agree with the comment. This is why we replaced the word and we are now using the word “theoretically”. We have also added the following including two new References: “However, the possibility of premature ovulation is rather unlikely because a follicle usually ruptures when it is mature, i.e. when it exceeds 16 mm in diameter in the presence of a fully expressed LH surge (Gougeon 1986*). In the case of premature LH surge, on the one hand the follicles usually have not fully matured and on the other hand the LH surge is markedly attenuated and even less than a third of the normal amplitude (Messinis and Templeton, 1987). Data in mice have shown that for the follicle to rupture the LH surge must be at least 85% of normal size (Peluso et al., 1990**)”.
*Gougeon, A. Dynamics of follicular growth in the human: a model from preliminary results. Hum. Reprod. 1986, 1, 81-87.
** Peluso, J.J. Role of the amplitude of the gonadotropin surge in the rat. Fertil. Steril. 1990, 53, 150-154.
- There is much focus in your discussion on premature luteinization being the possible negative effect of early LH surge. I agree that this won't be a concern with egg donor cycles, but premature ovulation will still be a risk for these cycles and may bear some discussion as well.
A: We agree. For this reason we have added an extra part. It now reads:” ation [21], the second appears to be theoretically necessary. Premature ovulation is rare, also being possible even with GnRH antagonist use, also met in progestine-primed ovarian stimulation and natural cycles IVF. The rationale behind its obligatory inhibition during an IVF cycle is that it impedes a woman's chance to reach her full oocyte yield potential, and subsequently the success of assisted reproduction.”
Reviewer 2 Report
Paper JCM jcm-2307756 by Messinis et al.
Ovarian stimulation with FSH alone versus FSH plus a GnRH antagonist for elective freezing in an oocyte donor/recipient programme: a protocol for a pilot multicenter observational study
1. Overall opinion
This paper aims to elaborate a study protocol to check if it is possible not to use any medication to prevent the luteinizing hormone (LH) surge during ovarian stimulation for oocyte donor cycles with freeze all. The paper concerns the protocol elaboration of a pilot study. They intend to organize a prospective observational study, with seven Greece centres. One major problem is the lack of intended randomisation in the protocol, which will not allow for a real conclusion since there is nothing on the allocation reasons to a group or the other. It is a pity that this study is elaborated as an observational one. No firm conclusion will be driven from the results, and this aspect is not discussed in the discussion section.
- Recommendations
Randomization is suggested to be added and explained in the protocol.
3. Detailed comments
a. Introduction
Introduction is ok and is based on recent literature.
b. Material and methods
The definition of the inclusion and non-inclusion criteria is correct. However, the allocation of donors to group 1 or 2 is never described, which bis a very weak point. The stimulation protocols are correctly described, as the recipients’ endometrium preparation. The outcome variables are also clearly defined, primary and secondary.
Statistical methods are ok.
c. Discussion:
This section is ok but needs also to discuss the fact that, apparently, no randomization is intended
Author Response
This paper aims to elaborate a study protocol to check if it is possible not to use any medication to prevent the luteinizing hormone (LH) surge during ovarian stimulation for oocyte donor cycles with freeze all. The paper concerns the protocol elaboration of a pilot study. They intend to organize a prospective observational study, with seven Greece centres. One major problem is the lack of intended randomisation in the protocol, which will not allow for a real conclusion since there is nothing on the allocation reasons to a group or the other. It is a pity that this study is elaborated as an observational one. No firm conclusion will be driven from the results, and this aspect is not discussed in the discussion section.
A: Thank you for your time spent to carefully assess our paper and the valuable comments.
- Recommendations
Randomization is suggested to be added and explained in the protocol.
A: Thank you for your comment. This is not an RCT, where meticulous randomization process is obligatory. This is an observational cohort study. To further clarify allocation in the two arms, some sentences have been added. It now reads:” The allocation of patients will be achieved using a 1:1 proportional pattern, depending on the random order in which they enter each Unit. Notably, both patients and gynecologists will be aware of the study protocol used.”.
- Detailed comments
- Introduction
Introduction is ok and is based on recent literature.
- Material and methods
The definition of the inclusion and non-inclusion criteria is correct. However, the allocation of donors to group 1 or 2 is never described, which bis a very weak point. The stimulation protocols are correctly described, as the recipients’ endometrium preparation. The outcome variables are also clearly defined, primary and secondary.
Statistical methods are ok.
A: Thank you for your comment. This is not an RCT, where meticulous randomization process is obligatory. This is an observational cohort study. To further clarify allocation in the two arms, some sentences have been added. It now reads:” The allocation of patients will be achieved using a 1:1 proportional pattern, depending on the random order in which they enter each Unit. Notably, both patients and gynecologists will be aware of the study protocol used.”.
- Discussion:
This section is ok but needs also to discuss the fact that, apparently, no randomization is intended.
A: We really thank you for addressing this point. We have added a final paragraph in our discussion section, clearly dealing with this. It now reads:” Finally, we admit that this is an observational study that will carry the associated biases a priori; as a result, no definite conclusion can be drawn. In contrast, we believe that this is a new theory that has to be proven step-by-step. We fully acknowledge the anxiety of both patients and clinicians for premature LH surge and ovulation; especially for the later, we came across a lot of negative beliefs and fear, when we tried to recruit IVF centres. Our further – and initial- aim though, remains to conduct a proper RCT, to prove or disprove this theory, but with some preliminary data on our hands.”.